# Glycosides for Peripheral Neuropathic Pain: A Potential Medicinal Components

**DOI:** 10.3390/molecules27010255

**Published:** 2021-12-31

**Authors:** Miao-Miao Tian, Yu-Xiang Li, Shan Liu, Chun-Hao Zhu, Xiao-Bing Lan, Juan Du, Lin Ma, Jia-Mei Yang, Ping Zheng, Jian-Qiang Yu, Ning Liu

**Affiliations:** 1Department of Pharmacology, School of Pharmacy, Ningxia Medical University, 1160 Shengli Street, Yinchuan 750004, China; t_tmiaomiao@163.com (M.-M.T.); daliushanzi12@163.com (S.L.); zchdegree@163.com (C.-H.Z.); lanxbing@163.com (X.-B.L.); dujuand@163.com (J.D.); malinzill@sina.com (L.M.); m15809581390@163.com (J.-M.Y.); 2College of Nursing, Ningxia Medical University, 1160 Shengli Street, Yinchuan 750004, China; li_yuxiang@163.com; 3Ningxia Special Traditional Medicine Modern Engineering Research Center and Collaborative Innovation Center, Ningxia Medical University, 1160 Shengli Street, Yinchuan 750004, China

**Keywords:** neuropathic pain, epidemiology, glycosides, oxidative stress, transcriptional regulation, ion channels, membrane receptors

## Abstract

Neuropathic pain is a refractory disease that occurs across the world and pharmacotherapy has limited efficacy and/or safety. This disease imposes a significant burden on both the somatic and mental health of patients; indeed, some patients have referred to neuropathic pain as being ‘worse than death’. The pharmacological agents that are used to treat neuropathic pain at present can produce mild effects in certain patients, and induce many adverse reactions, such as sedation, dizziness, vomiting, and peripheral oedema. Therefore, there is an urgent need to discover novel drugs that are safer and more effective. Natural compounds from medical plants have become potential sources of analgesics, and evidence has shown that glycosides alleviated neuropathic pain via regulating oxidative stress, transcriptional regulation, ion channels, membrane receptors and so on. In this review, we summarize the epidemiology of neuropathic pain and the existing therapeutic drugs used for disease prevention and treatment. We also demonstrate how glycosides exhibit an antinociceptive effect on neuropathic pain in laboratory research and describe the antinociceptive mechanisms involved to facilitate the discovery of new drugs to improve the quality of life of patients experiencing neuropathic pain.

## 1. Introduction

Neuropathic pain is defined by the International Association for the Study of Pain (IASP) as pain arising as a direct consequence of a lesion or disease affecting the somatosensory system [1]. Following injury, the structure and function of the somatosensory nervous system undergoes a variety of chronic and pathological changes that will eventually lead to the pathological amplification of nociceptive and non-noxious stimulus signals. Collectively, these events lead to neuropathic pain, and this is usually accompanied by anatomical and physiological changes in the central nervous system (CNS) or peripheral nervous system (PNS) [2]. Gene mutations in the PNS may give rise to the alteration of receptors and ion channels that underlie certain rare neuropathic conditions such as erythromelalgia and paroxysmal extreme pain disorder [3].

The core process involved in the pathogenesis of neuropathic pain involves pathological changes in the somatosensory nervous system that lead to abnormal changes and disorders of sensory signals into the spinal cord and brain [4]. Besides nerve injury, a number of potential predisposing factors before and after the formation of a lesion can contribute to the manifestation of chronic pain, including age, gender difference, pain sensitivity, emotional disorders and cognitive impairment [5]. According to sensory classification, neuropathic pain can be divided into three types: ongoing pain, paraesthesias, and altered stimulus-response function, including allodynia (pain in response to a non-nociceptive stimulus), hyperalgesia (an increased pain sensitivity to a nociceptive stimulus), and loss of sensation in some areas. These factors predominantly manifest as burning pain, paroxysmal electric shock-like pain, and brush-evoked pain [6,7]. Trauma (surgery, amputation), metabolic disorders (diabetes, uremia), infection (herpes zoster, HIV), poisoning (chemotherapy), vascular disease (arteritis nodosa), and malnutrition are all considered to be factors related to the initiation of neuropathic pain.

A previous study reported that 17% of patients experiencing neuropathic pain in the UK considered their quality of life to be ‘worse than death’ [8]. As the global population ages, the incidence of neuropathic pain associated with diabetes and cancer survival will inevitably become more common [9]. Numerous patients suffer from moderate or severe pain for many years. It is very difficult to cure patients with such pain; only 40–60% of pain symptoms can be partially relieved [2,10]. The current analgesics that are used to treat neuropathic pain have emerged from neurology or psychiatry, such as antiepileptics (pregabalin) and tricyclic antidepressants (amitriptyline). And for antiepileptics, they are expected to be effective in the treatment of neuropathic pain since epilepsy and neuropathic pain share the same pathophysiology, namely neural hyperexcitability; in addition, drug repurposing is a growing practice in human medicine. Unfortunately, these drugs are often ineffective or induce intolerable side effects or withdrawal reactions [11]. Thus, we need to look for safer and more effective drugs, and it is convenient to access natural products compared with synthetic drugs, owing to the diversity of functional secondary metabolites. Natural products, especially plant secondary metabolites are important options for treating different pain conditions, such as alkaloids, flavonoids and glycosides [2,12,13,14]. As for glycosides, they are ubiquitous natural products, and due to their structural diversity, especially the aglycone part, they have numerous pharmacological activities, such as antinociceptive, neuroprotective, antioxidant and anti-inflammatory effects [15,16]. Evidence also suggests that glycosides can achieve a positive antinociceptive effect on neuropathic pain [17].

In this review, we selected 11 glycosides with more clear mechanisms and summarized their antinociceptive effects on neuropathic pain. The aim of this review is to provide a theoretical basis for a preclinical study for drug research and development, and then facilitate the discovery of drugs for the treatment of neuropathic pain.

## 2. Epidemiology of Neuropathic Pain

According to a recent systematic review and meta-analysis, approximately 18% of the general population in developing countries experience chronic pain [18]. In addition to adults, chronic pain is known to affect 11–38% of adolescents and approximately 3–5% of these patients report severe disability from it [19,20,21].

Due to the lack of diagnostic criteria for neuropathic pain in epidemiological studies at the population level, it is difficult to evaluate the true prevalence and/or incidence of it [9]. Moreover, most previous studies have focused on certain etiological conditions, such as trigeminal neuralgia, painful diabetic peripheral neuropathy, postherpetic neuralgia, cancer, and viruses [22,23]. Studies have shown that the incidence of neuropathic pain is related to gender, age, education, employment, the inhabited environment, and the socioeconomic level. Moreover, the incidence of pain is high in women, elderly people, those with less formal education, physical laborers, rural residents, and people of a low socio-economic level [19,24,25]. The prevalence of neuropathic pain in patients with trigeminal neuralgia is three times higher in women than in men (ranging from 0.03–0.3%) and has a significant impact on people aged 37–67 years [22].

A recent systematic review reported that the best estimate for the prevalence of neuropathic pain may be 6.9–10% in the general population [23]. In the U.S.A, one third of Americans suffer from chronic pain; this exceeds the sum of cases afflicted by heart disease, cancer, and diabetes. Moreover, approximately one fifth of patients with chronic pain are considered to have neuropathic pain [26,27]. The prevalence of neuropathic pain in the U.S.A. is similar to that in other industrialized countries; the prevalence in Canada and the UK was reported to be 7.7% [28] and 8% [26], respectively. Neuropathic pain is often associated with a specific condition, such as diabetes, traumatic brachial plexus injury, bone metastases, sickle cell disease, and advanced cancer; the prevalence of neuropathic pain for these diseases has been reported to be 50%, 56%, 25.8%, 20–25% and 30.6%, respectively [29,30,31,32,33]. It is important that clinicians consider the existence of neuropathic pain characteristics when treating patients so as to optimize patient care and the management of symptoms. 

Recent research shows that one of the most common symptoms of coronavirus disease 2019 (COVID-19) is pain [34]. Patients with chronic pain who are infected with severe acute respiratory syndrome coronavirus 2 (SARS-CoV-2) might experience the aggravation of symptoms or the deterioration of the nervous system [35]. Neurological complications that are associated with coronaviruses (for example, Guillain-Barre syndrome and chronic inflammatory demyelinating polyneuropathy) are at risk of developing peripheral neuropathic pain. SARS-CoV-2 often invades the nervous system directly or via post-viral immune reactions and then causes peripheral or central neurological complications. As a consequence, a patient may develop COVID-19-induced neuropathic pain due to neurological complications [36]. As many as 2.3% of inpatients with COVID-19 have reported potential neuropathic pain. In fact, since chronic pain usually occurs within a few months after nerve injury, the true prevalence may easily be underestimated [35]. 

## 3. Pharmacological Treatment of Neuropathic Pain

At present, the main clinical modality used to treat neuropathic pain is drug therapy [37]. Finnerup et al. carried out a systematic review and meta-analysis for the pharmacotherapy of neuropathic pain by considering the Grading of Recommendations Assessment, Development, and Evaluation (GRADE) guidelines and recommendations put forward by the Special Interest Group on Neuropathic Pain (NeuPSIG). This analysis revealed that tricyclic antidepressants, serotonin-noradrenaline reuptake inhibitors, pregabalin, and gabapentin were strongly recommended as the first-line treatment for neuropathic pain. Weaker recommendations were also reported for a range of second-line drugs, including lidocaine patches, high concentration capsaicin patches, and tramadol. A number of third-line drugs were also recommended, including strong opioids and botulinum toxin A, although topical agents and botulinum toxin A were only recommended for peripheral neuropathic pain. This study also provided strong recommendations against the use of levetiracetam and mexiletine [38] (Table 1).

Boyle et al. explored the antinociceptive effect of amitriptyline, pregabalin, and duloxetine on patients with painful diabetic neuropathy; analyses showed that all of the medications tested reduced pain when compared with placebo, but no one treatment was superior to any other. Subjective pain ratings (BPI severity) showed improvements of approximately 50%; this was in line with previous research [39]. Another study showed that in patients with localized neuropathic pain (LNP) who treated with a 5% lidocaine patch and placebo for three months, dynamic mechanical allodynia diminished progressively by ≥30% in 96% of patients and by ≥50% in 83% of patients, and cold pain and maximal mechanical pain thresholds improved [40]. Although some drugs have targeted mechanism research, there are still some analgesics which are symptomatic and lack an in-depth study of underlying pain mechanisms. Consequently, these drugs provide a brief respite from pain and are only effective for a few patients. Generally, most of these drugs exhibit side effects that limit their clinical application [39], such as dizziness [41], cardiotoxicity [42], erythema [42], addiction [43], and respiratory depression [44]. Therefore, the research and development of new drugs against neuropathic pain that act in a more targeted and safer manner has become a significant hotspot in pain research. 

Natural products are important resources for the development of innovative drugs. Statistical analysis shows that only 36% of the 1073 new chemical molecular entities produced between 1981 and 2010 involved pure synthesis; rather, more than half came from natural products. Many compounds were found in higher plants between January 1981 and September 2006; indeed, 28% of new drugs were derived from natural products and their derivatives. Of these, natural compounds from plants have become indispensable modern drugs [45]. For example, vincristine is an anticancer drug derived from *Catharanthus roseus* (L.) G. Don [46,47], while artemisinin is an important antimalarial drug and a potential anticancer drug that is derived from *Artemisia annua* L. [48]. The multi-component and multi-target characteristics of medicinal plants can significantly improve their therapeutic effects; it is more convenient and efficient for medicinal plants components in the process of drug discovery and screening compared with synthetic drugs; more and more researchers are now focusing on the efficacy of plant components extracted from various traditional medical systems, such as alkaloids, flavonoids and glycosides; Therefore, researchers tend to look for components with a antinociceptive effect on neuropathic pain from natural products [7,17,49]. 

## 4. Synopsis of Glycosides

Glycosides are formed by connections between saccharide groups and non-saccharide substances via the terminal carbon atoms of sugar. Glycosides are the secondary metabolites of plants and are associated with a bitter taste [50]. Further modification of the glycosides (for example, by acylation, oxidation, and degradation) has further expanded the structural and functional diversity of metabolites [51]. Structurally, most glycosides are acetals that are formed by dehydration and condensation of the hemiacetal hydroxyl group of saccharides and the hydroxyl group of aglycone (the non-sugar portion). The formation of a bond that connects aglycone to a sugar atom is referred to as a glycosidic bond. Differences between glycoside bonds allow us to divide the glycosides into *N*-glycoside, *O*-glycoside (which is the most common and abundant), *S*-glycoside, and *C*-glycoside. Although some glycosides are simply linked with sugars to increase their transport in vivo, some large glycosides feature aglycones that are modified by complex oligosaccharides; these have special functions that are generated by the synergistic effect of aglycones and sugars [52] (Table 2). Glycosides are widely distributed in nature and are the active ingredients of many Chinese herbal medicines. Furthermore, glycosides can be distributed in all parts of plant anatomy, especially the roots and rhizomes. For example, *Panax notoginseng* (Burkill) F. H. Chen ex C. H. is known to have the highest content of saponins in its roots and rhizomes [53]. Glycosides have multiple pharmacological activities; for example, vitexin is one of the main components of hawthorn and exerts antinociceptive, anti-tumor, antihypertensive, anti-inflammatory, and antispasmodic effects [54]. Saikosaponin is the main chemical component of Radix Bupleuri and exerts effects against neuropathic pain, along with anti-inflammatory, immunomodulatory, antibacterial, and antiviral effects [55]. In one hand, glycosides’ multiple pharmacological activities related to their structure-activity relationships [56]. For antinociceptive activity, removal of hydroxyl groups from glycoside ligands, hydroxyl methylation, elimination of phenylacryloyl or phenylethyl will reduce the antinociceptive effect, and antioxidant activity is determined by the number of unsaturated bonds, the number and location of phenolic hydroxyl groups and carbon chain length [57]. In another hand, glycosides have the characteristics of multiple targets and multiple mechanisms, including the paeoniflorin regulating p38 MAPK/NF-κB signaling pathway and anti-insulin resistance via LKB1/AMPK and AKT pathways [58,59]; salidroside has an antinociceptive effect via regulating the activation of P2X7 receptors and has neuroprotective effects on the PI3K/Akt/TERT signaling pathway [60,61]. But whether glycosides’ antinociceptive effect can dominate other pharmacological effects remain no article has been reported yet.

Many glycosides have an antinociceptive effect; in this review, there are 11 glycosides that are active components of traditional Chinese herbs that have been mentioned because of their clear pharmacological mechanism or significant characteristics (such as local analgesia) or unique chemical structure. 

## 5. The Antinociceptive Effect of Glycosides in a Rodent Model of Neuropathic Pain

Clinically, patients with neuropathic pain usually report mechanical and thermal hypersensitivity which can be easily be detected [62]. And the characteristic manifestations of peripheral neuropathic pain in the animal model are basically consistent with the clinical manifestations. The loss of myelinated Aα/β and Aδ fibers induced by the Wallerian degeneration was shown to produce ipsilateral reflex hyper-responsiveness to mechanical and thermal stimuli in neuropathic pain conditions; Aβ-fibers-mediated mechanical hypersensitivity (touch-pressure), C-fibers-mediated thermal hypersensitivity (hot pain) [63].

Allodynia and hyperalgesia are the characteristic manifestations of peripheral neuropathic pain which are the symptoms and signs that may serve as readouts for pain and then contribute to the improved delineation of neuropathic pain, thus providing measures for the structures within the nervous system where signs of neuronal hyperexcitability are present [64]. Clinically meaningful pain reduction is ≥30% reduction in pain from baseline [65]. Allodynia refers to pain in response to a non-nociceptive stimulus; it should only be used when it is known that the test stimulus is not capable of activating nociceptors, such as brush the skin; hyperalgesia refers to increased pain sensitivity, it may include both a decrease in threshold and an increase in suprathreshold response, when we cannot confirm whether or not the test stimulus is capable of acting nociceptors, it is useful to have an umbrella term (hyperalgesia) for all types of increased pain sensitivity [66].

Over the years, a number of laboratory studies have reported associations between glycosides and antinociceptive effect [17]. In the next section of this review, we describe the antinociceptive effect of glycosides on neuropathic pain (Table 3).

### 5.1. Hesperidin

Hesperidin is isolated from *Citrus reticulata* Blanco [83]. In recent studies, hesperidin can both applicable in chronic constriction injury (CCI) and streptozotocin (STZ)- induced neuropathic pain rats. Administration of hesperidin (50 mg/kg or 100 mg/kg) resulted in a relief of mechanical hyperalgesia and thermal hyperalgesia in a CCI-induced neuropathic pain model of rats [70,71]. In another study, hesperidin (25, 50, and 100 mg/kg) was administered to STZ-induced painful diabetic neuropathy rats for four weeks, then lead to a respite in mechanical allodynia and thermal hyperalgesia [72].

To sum up, in addition to relieving neuropathic pain caused by CCI, hesperidin can relieve pain while alleviating the symptoms of diabetes in STZ-induced painful diabetic neuropathy rats; it can indicate improving blood sugar, alleviate nerve injury, and it also shows that the action of the drug is diverse.

### 5.2. Saikosaponins

Saikosaponins are isolated from *Bupleurum chinense* DC. [84]. A previous study found that the continuous administration of saikosaponin A (6.25, 12.5 and 25 mg/kg) over 14 days alleviated mechanical allodynia and thermal hyperalgesia in CCI-induced neuropathic pain rats in a dose-dependent manner [55].

Saikosaponin D (5, 20 mg/kg) was also shown to significantly attenuate agonist-induced nociceptive responses and vincristine-induced mechanical hypersensitivity in neuropathic pain mice [73].

However, researchers thought saikosaponin A alleviated neuropathic pain in a dose-dependent manner via three doses may inappropriate, it can be described as a dose-effect relationship, whether saikosaponin A alleviates neuropathic pain in a dose-dependent manner needs continued study.

### 5.3. Isoorientin

Isoorientin is isolated from *Pueraria lobata (Willd.) Ohw i* [85], *Phyllostachys heterocycla (Carr.)*, *Mitford cv. Pubescens* [86], and *Gentiana scabra* Bunge [87]. Researchers have shown that the intragastric administration of isoorientin (7.5, 15 and 30 mg/kg) over eight days alleviated CCI-induced mechanical allodynia, cold allodynia, and thermal hyperalgesia of mice [61]. These authors also observed that the administration of isoorientin also relieved tissue damage such as disordered myelinated nerve fibers, swollen axons, neuron gaps, and unmyelinated axons of the sciatic nerve trunk, as determined by hematoxylin-eosin (H&E) staining and transmission electron microscopy [80].

As usual, there are a variety of experimental methods to explore the antinociceptive effect of drugs on neuropathic pain. In addition to behavioral experiments, there are histopathological staining experiments to study the neuroprotective effect of drugs, such as H&E staining to observe the relief of sciatic nerve fiber injury (nerve fiber rupture and neuron vacuole and so on) after administration, and transmission electron microscopy to observe the submicroscopic structure of nerve fibers and understand the changes of the myelin sheath.

### 5.4. Liquiritin

Liquiritin is isolated from the roots of *Glycyrrhiza uralensis* Fish [88,89]. A previous study showed that CCI-induced neuropathic pain mice were administration of liquiritin (30, 60, and 120 mg/kg, i.p.) for 8 consecutive days, meanwhile, mice were tested from 7–14 days post-surgery with the Von Frey Filament test, the cold-plate test, and the radiant heat test, then researchers observed that liquiritin could reduce mechanical allodynia, thermal hyperalgesia and cold allodynia [78]. 

In addition to the analgesic effect of glycyrrhizin on CCI-induced neuropathic pain, recent studies have reported that liquiritin can inhibit SARS-CoV-2 by mimicking type 1 interferon [90]; thus, perhaps liquiritin can alleviate the symptoms of neuropathic pain caused by COVID-19.

### 5.5. Albiflorin

Albiflorin is isolated from *Radix paeoniae* Alba [91]. Albiflorin is transformed to benzoic acid in the intestine, a vital metabolite that can cross the blood-brain barrier, improve brain function, and exhibit antidepressant activity [92]. The administration of albiflorin 50 mg/kg and MCC950 (a selective inhibitor of the NOD-like receptor protein 3 (NLRP3) inflammasome) at a dose of 10 mg/kg for 15 days led to a number of improvements thereafter, including the alleviation of CCI-induced mechanical allodynia, comorbid anxiety, and depression-like behavior, as determined by the Von Frey test, the open field test, the elevated plus maze test, and the forced swim test [81].

Neuropathic pain is usually accompanied by anxiety and depression, so drugs usually require not only antinociceptive effect but anti-anxiety and anti-depressant activities, and albiflorin just meets the above conditions, so it may be a promising analgesic. 

### 5.6. Salidroside

Salidroside is the main active component extracted from *Rhodiola rosea* L. [60]. A previous study investigated a model of painful diabetic neuropathy, the Zucker diabetic fatty (ZDF) rat which widely used as a model for human diabetes and its complications, including neuropathic pain were given salidroside (25, 50, and 100 mg/kg) and then observed, and the authors found that the administration of salidroside (25 mg/kg and 50 mg/kg) showed a partial recovery from allodynia during the 5th week of the experiment and relieved thermal hyperalgesia by the fifth and eighth week, respectively; however, there were no significant improvements in the 100 mg/kg group [61,93,94]. Another study administered salidroside (50 mg/kg/day and 100 mg/kg/day) to STZ-induced type 2 diabetic rats (that had been fed on a HFD for 4 weeks) for six weeks; analysis showed that both mechanical allodynia and thermal hyperalgesia were alleviated. Analysis also showed that salidroside relieved type 2 diabetes mellitus (T2DM) by improving hyperglycemia and by ameliorating insulin resistance [79]; these results were critical because T2DM is characterized by hyperglycemia and insulin resistance [95].

Interestingly, models of painful diabetic neuropathy, salidroside at a dose of 100 mg/kg did not have significant antinociceptive effect to neuropathic pain in ZDF rats, but salidroside 100 mg/kg could relieve STZ-induced mechanical allodynia and thermal hyperalgesia; the reasons for this need to be explored further.

### 5.7. Morroniside

Morroniside is isolated from the dry ripe sarcocarp of *Cornus officinalis* Sieb. et Zucc. [96], and the leaves of *Sarracenia purpurea* L. [97]. Morroniside (30, 100, 300, 600 mg/kg, p.o. or 3, 10, 30, 100, 300 μg i.t.) has been shown to significantly relieve mechanical allodynia and thermal hyperalgesia in spinal nerve ligation (SNL)- induced neuropathic pain in rats in a time- and dose-dependent manner [76,77]. Furthermore, the rotarod test was used to demonstrate that the antinociceptive effect of morroniside was not due to a false positive reaction caused by sedation and had no adverse effect on motor function. To test the analgesic tolerance of morroniside, repeated daily intrathecal injections of morroniside (100 μg) in SNL-induced neuropathic pain in rats for seven days increased paw withdrawal thresholds in the ipsilateral paws by 58% and the significant mechanical antiallodynic effect could persisted for the seven days [76].

Morroniside has a significant antinociceptive effect on neuropathic pain; repeated daily injections of morroniside did not induce antinociceptive tolerance, which is its significant advantage, so it is a potential and promising compound for neuropathic pain.

### 5.8. Verbascoside

Verbascoside is extracted from the leaves of *Penstemon barbatus* (Cav.) Roth [98], *Lippia citriodora* H.B.K. [99], and *Paulownia tomentosa* Steud [100]. Mass spectrometry further showed that caffeic acid was the main metabolite of verbascoside. Verbascoside is known to be apower scavenger for reactive oxygen species (ROS) and can also act as an antioxidant agent [101]. In a previous study, Verbascoside (50, 100 and 200 mg/kg) was administered intraperitoneally for 14 days continuously from the day of surgery in CCI-induced neuropathic pain rats could alleviate mechanical allodynia, cold allodynia, and heat hyperalgesia, verbascoside was shown to significantly attenuate these behavioral changes associated with neuropathy [69]. In another study, Isacchi et al. found that verbascoside (100, 300, 600 mg/kg) reversed mechanical hyperalgesia in CCI- or intra-articular injection of sodium monoiodoacetate (MIA)-induced neuropathic pain rats as determined by the paw pressure test [99]. 

Verbascoside has a significant antinociceptive effect, but higher water solubility of verbascoside could lead to low bioavailability of this compound. Derivatives of verbascoside and new formulations with more penetration to lipophilic vehicles could help to reduce the effective doses of this agent.

### 5.9. Paeoniflorin

Paeoniflorin is a major active compound isolated from the root of *Paeonia lactiflora* Pall [102,103,104]. In PTX (paclitaxel)-induced neuropathic pain mice, Andoh et al. administered a single intraperitoneal injection of PTX followed by the topical administration of paeoniflorin (0.1% and 1%; 20 μL) to the left or both planar surfaces twice daily for 13 days, then observed that the PTX-induced mechanical allodynia was significantly inhibited, but a single administration of paeoniflorin was invalid. They also demonstrated that paeoniflorin exhibited local inhibitory activity. Paeoniflorin (1%) was applied repeatedly on left plantar surfaces for 13 days after PTX injection; the authors reported that the mechanical allodynia was only relieved in the treated paw [74]. Other studies, investigating CCI-induced neuropathic pain in rats, administered paeoniflorin (25, 50, 100 mg/kg) for 11 days continuously after surgery, and ethological analyses demonstrated that paeoniflorin significantly increased the mechanical withdrawal threshold and thermal withdrawal latency on days one, three, seven, and 11 [58,75]. 

The local antinociceptive effect of paeoniflorin is a significant characteristic, as generally the adverse reactions of local medication are less than those of systemic medication. Thus, if paeoniflorin can enter the market, it may have significant development prospects.

### 5.10. Diosmin

Diosmin (diosmetin-7-O-rutinoside) is isolated from *Scrophularia nodosa* and can be obtained via hesperidin dehydrogenation [105]. In CCI-induced neuropathic pain rats, diosmin (10, 100, 316.2, 562.3, and 1000 mg/kg) could attenuate mechanical and thermal hyperalgesia in acute treatments (0–120 min) except 10 mg/kg, and diosmin (316.2 mg/kg) could also attenuate mechanical and thermal hyperalgesia in sub-chronic (15–28 days) treatments, ultra-high-performance liquid chromatography tandem mass spectrometry (UHPLC-MS) analysis further found that diosmin could cross the blood brain barrier [67]. Another study showed that mice withCCI-induced neuropathic painthat received a single administrated diosmin (1, 10 mg/kg) at the seventh day or received prolonged administration of diosmin (10 mg/kg) at the 7th–14th days exhibited inhibited mechanical and thermal hyperalgesia, and they also demonstrated long term application of diosmin do not causing gastric leision or renal damage [68]. 

In conclusion, long-term and short-term administration of diosmin can alleviate the symptoms of neuropathic pain, at the same time, it also shows that long-term administration of diosmin may have no tolerance.

### 5.11. Geniposide

Geniposide is isolated from *Gardenia jasminoides Ellis* [106]. Studies show that geniposide produces antinociception during persistent pain [107]. In a streptozotocin (STZ)-induced painful diabetic neuropathy model of SD rats, geniposide (1, 10 and 100 mg/kg) was administered intraperitoneally for seven days continuously from the 21st day after modeling, then researchers found geniposide could alleviate mechanical hyperalgesia in addition to reducing blood glucose in diabetic neuropathy SD rats [82]. 

Studies have shown that the iridoid structure represented by geniposide analogs is a novel core structure for activating glucagon-like peptide-1 receptors (GLP-1Rs, recently discovered pain receptors); thus, the study of geniposide may lay a foundation for the development of new neuropathic pain analgesics targeting GLP-1R.

Collectively, these eleven glycosides have significant antinociceptive effects in preclinical pharmacological experiments, and have potential as therapeutic agents for neuropathic pain.

## 6. Antinociceptive Mechanisms Underlying the Action of Glycosides on Neuropathic Pain

Mechanism-based treatment is closely related to disease/cause-based treatments. By knowing how certain diseases (e.g., diabetes) or other factors cause (e.g., oxaliplatin) neuropathic pain, we can use a further mechanism-based approach to obtain optimal analgesics. A number of studies have shown that glycosides are involved in multiple peripheral mechanisms of neuropathic pain, as described in the following section.

### 6.1. Glycosides and Oxidative Stress

Oxidative stress and inflammation have been proposed to play both causative and deteriorative roles in neuropathic pain. Elevated levels of ROS have been detected at sites of inflammation and tissue injury [108]. Matrix metalloproteinases (MMPs) are responsible for neuropathic pain; previous research has shown that MMP-9 is correlated with oxidative stress and the activation of glial cells [109,110]. Peripheral nerve injury is known to elevate ROS levels; in turn, ROS activates MMP-9 and then activates cytokines, including IL-1β. These cytokines act on adjacent microglia and astrocytes, which then release pain-causing substances such as pro-inflammatory cytokines. Subsequently, these pro-inflammatory cytokines can activate activated glial cells and neurons via positive feedback loop mechanisms. Collectively, these processes enhance peripheral and central sensitization to maintain neuropathic pain [111,112,113]. In a model of CCI-induced neuropathic pain, the administration of isoorientin was shown to alleviate oxidative stress by influencing markers of oxidative stress, including total antioxidant capacity (T-AOC), catalase (CAT), and total superoxide dismutase (T-SOD). These changes increased malondialdehyde (MAD), thus leading to the expression of MMP-9; this significantly reduced oxidative stress and inflammation and the activation of microglia and astrocytes, thus reducing the expression levels of proinflammatory cytokines. Therefore, isoorentin plays an antinociceptive role in neuropathic pain by inhibiting oxidative stress, the activation of MMP-9, and neuroinflammation [80].

### 6.2. Glycosides and Transcriptional Regulation

#### 6.2.1. Glycosides and Pro-Inflammatory Cytokines

Over recent years, research has repeatedly emphasized the importance of the neuroinflammatory response in the development and maintenance of inflammatory and neuropathic pain. The inflammatory cells that cause this reaction include circulating immune cells, such as monocytes, T and B lymphocytes, neutrophils, and glia in the central nervous system. Pain signals are transmitted by sensory neurons in the peripheral nervous system. These neurons express different receptors and ion channels and respond to mediators secreted by these inflammatory cells [114]. Pro-inflammatory cytokines are involved in demyelination and the degeneration of peripheral nerves and can also increase the excitability of sensory afferents, thus inducing neuropathic pain [115].

In vivo, saikosaponin A can significantly reduce the levesl of TNF-α, IL-1β, and IL-2 protein expression, along with *p-p38* and *nuclear factor kappa B* (*NF-κB)* mRNA and protein expression levels in the spinal cord, so as to exert an antinociceptive effect on neuropathic pain [55]. Hesperidin, diosmin and liquiritin are also known to regulate pro-inflammatory cytokines (IL-1β, IL-6, and TNF-α) in a mouse model of CCI-induced neuropathic pain, thereby alleviating mechanical and thermal hyperalgesia [71,78]. In an STZ-induced model of painful diabetic neuropathy, the co-administration of hesperidin (100 mg/kg; per os) and insulin 10 IU/kg (subcutaneous injection) was shown to play a synergistic effect; the highest antinociceptive activity occurred by controlling hyperglycemia and downregulating hyperlipidemia to down-regulate the generation of free radicals, the release of pro-inflammatory cytokines, and the elevation of membrane bound enzymes to reverse mechanical and thermal hyperalgesia [72] (Figure 1).

#### 6.2.2. Glycosides and the NOD-Like Receptor Protein 3 (NLRP3) Inflammasome

NLRP3 inflammasome activity is a necessary component of the innate immune response to tissue damage. The dysregulation of inflammasome activity has been implicated in a number of neurological conditions [116]. The NLRP3 inflammasome is composed of NLRP3, adaptor ASC, and caspase-1 enzymes, and can be activated by inflammatory stimuli [116,117]. Recent studies have shown that the NLRP3 inflammasome also exists in the neurons of the sensory nervous system and are related to neuropathic pain [81,118,119].

In models of CCI-induced neuropathic pain, paeoniflorin was shown to inhibit NLRP3 and pro-IL-1β by inhibiting the regulation of NF-κB-mediated transcription. NLRP3 is the most extensive regulator of Caspase-1 activation and can produce and activate caspase-1, and then cleave pro-IL-1β into mature IL-1β. Paeoniflorin can also inhibit the downstream bioactive components caspase-1 and IL-1β, thus inhibiting neuropathic pain. However, after applying a selective inhibitor of NLRP3 (MCC950), the antinociceptive effect and related protein expression of paeoniflorin and MCC950 on neuropathic pain were very similar; this suggested that the antinociceptive effect of paeoniflorin on neuropathic pain occurred by inhibiting the activity of the NLRP3 inflammasome [75]. Glycosides do not only exhibit an antinociceptive effect; they can also alleviate mood disorders. It has been demonstrated that the NLRP3 inflammasome is involved in the pathological processes of anxiety and depression during neuropathic pain conditions. Albiflorin can relieve anxiety and depression-like behavior under neuropathic pain conditions, and these effects are regulated by the Kelch-like ECH associated protein 1 (Keap1)/nuclear factor erythroid 2-related factor 2 (Nrf2) signaling pathway; these processes reduced antioxidant levels to suppress the hippocampal activation of NLRP3 inflammation [81] (Figure 1).

### 6.3. Glycosides and Ion Channels

#### 6.3.1. Glycosides and P2X Receptors

Adenosine triphosphate (ATP) is a key modulator in nociception mechanisms [120]. P2X receptor 7 (P2X7R) is a plasma membrane receptor that exhibits the lowest sensitivity for extracellular ATP in the P2XR family [121,122]. This high threshold for activation confers a damage-sensing role to P2X7 as this receptor only triggers downstream effects when the concentration of ATP is pathologically elevated. P2X7R acts as a cell surface regulator for inflammasome recruitment and the synthesis and release of pro-inflammatory cytokines. Previous studies have also shown that multiple inflammatory cascades are involved in neuropathic pain [122,123]. The P2X receptor plays an important role in mediating ATP signals to produce and maintain chronic pain; thus, this receptor is considered to be a promising antinociceptive target.

Following the administration of salidroside (25 mg/kg and 50 mg/kg) for eight weeks, the levels of P2X7R, TNF-α, and IL-1β were significantly reduced in a rat model of painful diabetic neuropathy [61]. In a previous in vitro study, pEGFP-hP2X7 was transfected into HEK293 cells. Then, salidroside was used to inhibit ATP (100 μM)-activated currents, as monitored by whole cell patch clamping. These data indicated that salidroside could not only down- regulate the expression of the P2X7 receptor, but it could also directly target and inhibit the P2X7 receptor [61].

#### 6.3.2. Glycosides and Transient Receptor Potential Ankyrin 1 (TRPA1)

Members of the transient receptor potential (TRP) channel family are known to be a kind of ion channel and have attracted significant attention in the search for new therapies for neuropathic pain. Members of this broad class of ion channels are involved in a wide range of processes associated with sensory perception, including pernicious chemicals, along with mechanical and thermal sensations [124,125]. Transient receptor potential ankyrin 1 (TRPA1) is a member of the family of TRP channels; specifically, TRPA1 is a non-selective cation channel and is mainly expressed in the neurons of the root ganglion [126]. Studies have shown that the TRPA1 channel, as a major pain transducer, plays a key role in neuropathic pain (especially in the transition from acute pain to chronic pain), and can mediate mechanical hyperalgesia and cold hypersensitivity in the neuropathic pain caused by chemotherapy [127]. Research has demonstrated that cisplatin-induced mechanical hypersensitivity was inhibited by the silencing of TRPA1 expressing afferents [128].

In previous research, high-throughput screening (HPS) demonstrated that saikosaponins which contain glycosyl and aglycone components could inhibit TRPA1, so it is necessary to determine whether the glycosyl component of saikosaponins is required to inhibit TRPA1, then aglycone from saikosaponins such as saikogenins A and D were tested by calcium assays, and results showed that they had no effect on agonist-stimulated calcium flux, thus indicating that there is a requisite role for the active sugar group of saikosaponins [73]. AITC is an agonist of the TRPA1 channel in primary afferent neurons and is known to lead to an acute pain response in rodents. Following the generation of an AITC-induced pain model, researchers found that saikosaponin D (50 mg/kg) significantly reduces the nociceptive response time by approximately 25%, thus indicating that saikosaponin D could inhibit the activation of TRPA1 induced by AITC in vivo, thus proving that saikosaponin D acts as an antagonist of TRPA1. Studies have shown that TRPA1 is involved in chemotherapy-induced neuropathic pain [129]. Through the establishment of a vincristine-induced model of neuropathic pain, researchers found that saikosaponin D (20 mg/kg) could significantly alleviate mechanical hyperalgesia in mice and increase the mechanical withdrawal threshold to a level that was 80% of the normal group [73]. These findings demonstrated that saikosaponins exert an antinociceptive effect on chemotherapy-induced peripheral neuropathy by targeting TRPA1.

### 6.4. Glycosides and Membrane Receptors

Glycosides are orthosteric agonists of the G-protein-coupled receptor (GPCR) and act by the extracellular N-terminal ectodomain interacting with the C-terminal residues of cognate ligands, which positions the N-terminus of the ligand to interact with critical determinants in the receptor’s transmembrane region, thus leading to full activation and signal transduction. Previous studies have shown that a variety of GPCRs are involved in the antinociceptive process for neuropathic pain, including G-protein-coupled receptor 160 (GRP160) [130], G-protein-coupled receptor 34 (GPR34) [131], G-protein-coupled receptor 40 (GPR40) [132,133,134], and GLP-1R [77,135]. The class B G-protein-coupled GLP-1Rs are widely distributed in human islets, lungs, the cardiovascular system, the central nervous system, and other tissues. The main signal transduction pathways are calcium and the cAMP/PKA signal pathway [136,137]. Glycosides featuring a scaffold of cyclopentapyran with a double bond between C7 and C8 are reversible and orthosteric agonists of GLP-1R. The binding of GLP-1R to its ligand (GLP-1) leads to the pancreatic β cells releasing glucose-dependent insulin; therefore, GLP-1R is a key target for the treatment of type 2 diabetes [135,138,139,140,141]. Subsequently, researchers discovered that GLP-1R is a potential therapeutic target for a variety of chronic pain conditions and not just diabetes. The activation of GLP-1R, which is specifically expressed in the microglia of the spinal dorsal horn, subsequently results in the activation of the cAMP/PKA/p38β/CREB signaling pathway. Autocrine IL-10 then regulates the release of β-Endorphin from the microglia; β-Endorphin then mediates diverse inhibitory effects in hypersensitive pain states by acting on μ-receptors, rather than the antinociceptive effect produced by the anti-inflammatory activity of anti-inflammatory cytokines [135,142].

Morroniside is the major glycoside active constituent of *Cornus officinalis (C. officinalis)* Sieb. Et Zucc. and features a scaffold of cyclopentapyran with a double bond between C7 and C8. Studies have confirmed that the antinociceptive effect of morroniside on neuropathic pain is produced via the GLP-1R receptor. Morroniside is a reversible and orthosteric agonist of GLP-1R [143]. Exendin (9–39) is a specific antagonist of GLP-1Rs. In a previous study, thirty minutes after the intrathecal injection of saline (10 μL) or exendin (9–39) (2 μg), a rat model of neuropathic pain was orally administrated with morroniside (300 mg/kg) or given an intrathecal injection of morroniside (100μg). These researchers clearly observed mechanical anti-allodynia in the ipsilateral paws in the saline group. Conversely, the mechanical anti-allodynia effect of morroniside in the lesion was completely blocked by exendin (9–39), a GLP-1R antagonist [76]. Furthermore, Tang et al. studied the downstream mechanisms underlying the antinociceptive effect of morroniside on neuropathic pain via the activation of GLP-1R. Using a mouse model of neuropathic pain, researchers applied minocycline (a microglial inhibitor) or clodronate liposom (a microglial depletory agent) for 4 h or one day prior to morroniside treatment. This treatment reversed the antinociceptive effect of morroniside. However, the antinociceptive effect of morroniside can be reversed, thus indicating that the antinociceptive effect of morroniside occurs in microglia. Subsequently, the application of the IL-10 antibody, a β-Endorphin antibody, and a μ-Opioid receptor antagonist (CTAP) also showed that the antinociceptive effect of morroniside was reversed. Furthermore, molecular biological techniques showed that morroniside promotes the release of *IL-10* mRNA and *β-Endorphin precursor proopiomelanocortin (POMC)* mRNA in vivo and in vitro, thus verifying that morroniside produces therapeutic effects in neuropathy via the spinal microglial expression of IL-10 and the subsequent secretion of β-Endorphin after the activation of GLP-1R [77]. Likewise, geniposide is one of the main active components of the medicinal plant *Gardenia jasminoides Ellis*. Geniposide also features a scaffold of cyclopentapyran, with a double bond between C7 and C8; this produces antinociception during persistent pain by activating the spinal GLP-1Rs. Furthermore, the iridoids represented by geniposide are known to be orthosteric agonists of GLP-1Rs and exert similar functionality in humans and rats and presumably act at the same binding site as exendin (9–39) [107] (Figure 2).

## 7. Conclusions and Future Perspectives

In this review, a summary of the existing commonly used drugs helps to choose the appropriate treatment [37]. Understanding the mechanism of action of drugs is helpful to obtain optimal analgesics. The antinociceptive effect and mechanism of glycosides derived from medicinal plants are described, which facilitates mechanism-based identification of potential drug targets, and provides insights and ideas for the development of new drugs, playing an important role in the treatment of neuropathic pain.

Glycosides exert extensive biological activities and pharmacological effects. Aside from their antinociceptive effect on neuropathic pain, glycosides can also be used to treat a variety of diseases. Glycosides can be used to inhibit cancer [144], manage heart ailments [145], and alleviate atherosclerosis [146]. 

Although there are many glycosides, there is insufficient research relating to their antinociceptive effect for neuropathic pain. There are several reasons for this. First, poor oral bioavailability; in native forms the glycosides are not absorbed and have to be hydrolyzed by intestinal enzymes or by colonic microflora before absorption [56]. Second, glycosides are unstable, they are easily hydrolyzed into glycosyl and aglycone by acid, alkali and enzyme, so the prototype and metabolite which plays an antinociceptive role is in doubt [147]. Third, water solubility, poor fat solubility and weak membrane permeability limit glycoside penetration into the central nervous system. Fourth, the safety of glycosides is not completely clear. 

Studies have shown that protein-ligand complexes can be used to maintain the stability of bioactive components. At pH 6.3, a whey protein isolate was thermally modified at 85 ℃ to act as a carrier for compound cyanidin-3-o-glucoside (C3G) and was subsequently combined with small molecular antioxidants. The C3G residue within the protein-ligand complex reached a total of 36.8% after 30 days storage at 25 °C [148]. Currently, the most commonly used strategy to alter membrane permeability is the hydroxyl acylation of glycosides or the preparation of esters. For example, researchers have studied the acylation of salidroside and found that synthesized salidroside acrylate derivatives exhibit a certain neuroprotective effect [149]. Chemical modifications can increase their potency. Collectively, these data suggest that glycoside analgesia is a promising strategy for the future [99].

In this review, we describe multiple mechanisms, such as oxidative stress, pro-inflammation, ion channels, and membrane receptors; these mechanisms are all associated with neuropathic pain. Mitochondria are the main intracellular source of production for ATP, ROS, and reactive nitrogen species (RNS). The mitochondria are also involved in many other cellular functions including Ca^2+^ signaling and apoptosis [150]. Multiple TRP channel subtypes (such as TRPM2, TRPM4, TRPM3, TRPC3, TRPC4, TRPV1, TRPV4, and TRPA1) are targeted by oxidative stress and a variety of by-products. TRPA1 is the main sensor of oxidative stress and nitrative stress associated with pain [151,152,153]. The sensitization of TRPA1 by Ca^2+^, ROS, RNS, and their by-products reactive carbonyl species (RCS) could lead to a prolonged period of hypersensitivity to mechanical and cold stimuli. On the other hand, TRPA1 produces and releases neuropeptide substance P (SP) and calcitonin gene-related peptide (CGRP), thus resulting in neurogenic inflammation [108,154]. ROS activates NLRP3 and ASC in the NLRP3 inflammasome to increase the release of IL-1β. This process promotes neuropathic pain and activates MMP-9, thus causing glial cells to activate and release many pro-inflammation factors, thus resulting in neuro-inflammation [75,80]. ATP produced by the mitochondria activates P2X and P2Y purinergic receptors. P2X receptors are trimeric ATP-gated cation channels that are permeable to sodium, potassium, and calcium. These channels can open within milliseconds of ATP-binding [155,156]. Using extracellular ATP as a pro-inflammatory molecule, P2X receptors can trigger inflammation in response to high ATP release, thus causing neuropathic pain [157]. However, with respect to the functional ability of glycosides, there are only a few TRP channel subtypes. P2 purinergic receptors have been investigated due to their association with neuropathic pain (e.g., TRPA1 and the P2X7 receptor). It is important that we investigate interactions between multiple mechanisms in more detail in the future.

The main purpose of drug research is for the drug to go to market. For glycosides, the following steps are generally adopted: the establishment of a biological hypothesis of the mechanisms involved in the pathology of pain; the identification of a potential molecular target based on the biologic hypothesis to screen for new analgesic glycosides with higher potency and better safety; the optimization of its structure and the testing in animal models; and the preparation for a three-phase clinical trial. If all the above steps are carried out smoothly, then the glycosides which were selected can be used clinically.

## Figures and Tables

**Figure 1 molecules-27-00255-f001:**
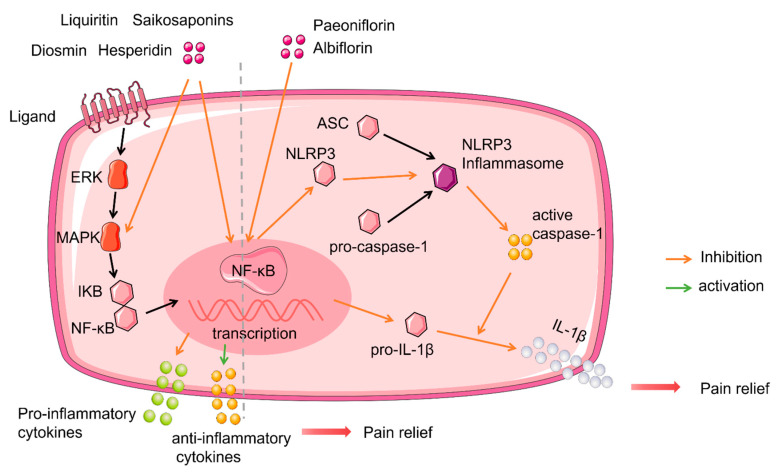
Glycosides and transcriptional regulation. Liquitin, saikosaponins, diosmin and hesperidin produced relief of neuropathic pain via inhibited NF-κB transcription downregulated pro-inflammatory cytokines, upregulated anti- inflammatory cytokines. Paeoniflorin and albiflorin inhibited NF-κB transcription and downregulated the transcription levels of inactive NLRP3 and pro-IL-1β contribute to the reduce of NLRP3 inflammasome synthesis, then active caspase-1 cleaved pro-IL-1β, reduced IL-1β released, lead to pain relief.

**Figure 2 molecules-27-00255-f002:**
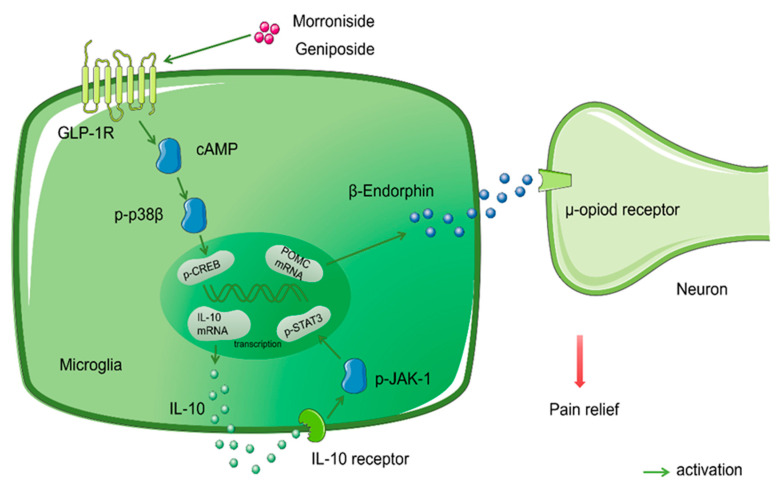
Glycosides and membrane receptors. Morroniside and geniposide activated GLP-1R (one of GPCRs), via IL-10/β-Endorphin signaling pathways mediated analgesia in neuropathic pain.

**Table 1 molecules-27-00255-t001:** Recommended drugs for the treatment of neuropathic pain based on the GRADE classification.

Grade	Drug	Neuropathic Pain Conditions	Major Side-Effects	Strength of Recommendation
First-line drug				
	Serotonin-noradrenaline reuptake inhibitors duloxetine and venlafaxine	All	Nausea	Strong
	Tricyclic antidepressants	All	Sedation, anticholinergic effects	Strong
	Pregabalin, gabapentin, gabapentin extended release or enacarbil	All	Sedation, dizziness, peripheral oedema	Strong
Second-line drug				
	Tramadol	All	Nausea/vomiting, constipation, dizziness	Weak
	Capsaicin 8% patches	Peripheral	local pain, oedema, and erythema	Weak
	Lidocaine patches	Peripheral	Local erythema, rash	Weak
Third-line drug				
	Strong opioids	All	Nausea/vomiting, constipation, dizziness	Weak
	Botulinum toxin A	Peripheral	local pain	Weak

**Table 2 molecules-27-00255-t002:** Structure of several glycosides.

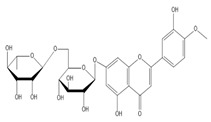	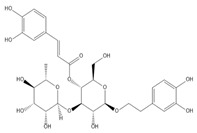	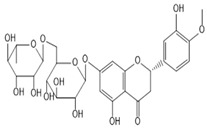	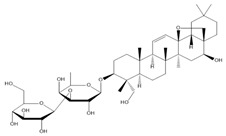
Diosmin	Verbascoside	Hesperidin	Saikosaponin D
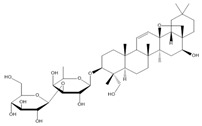	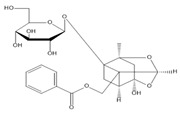	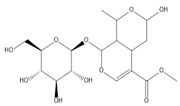	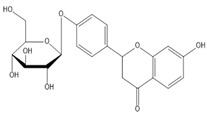
Saikosaponin A	Paeoniflorin	Morroniside	Liquiritin
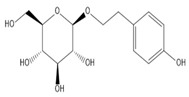	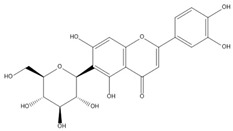	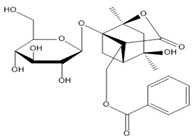	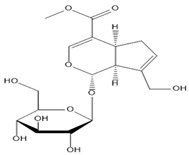
Salidroside	Isoorientin	Albiflorin	Geniposide

**Table 3 molecules-27-00255-t003:** Pharmacological activities, behavioral experiment results and mechanisms of glycosides.

Name	Pharmacological Activities	Animals	Model	Dose mg/kg (Route of Administration)	Nociceptive Tests														Mechanism	Refences
					Randall–Selitto Paw Pressure Test	Von Frey Filament Test	Radiant Heat Test	Hot Plate Test	Tail Immersion Test	Acetone Drop Test	Cold Plate Test	Rotarod Test	Spontaneous Exploratory Test	Walking Test	Open-Field Test	Elevated Plus Maze Test	Forced Swim Test	Electrophysiology Examination		
Diosmin	Antinociceptive, anti-inflammatory	Male Wistar rats	CCI	10, 100, 316.2, 562.3 or 1000 mg/kg, i.p.	-	↓ Mechanical hyperalgesia	↓ Thermal hyperalgesia		-		-				→ Locomotor activity			-	↓ pro-inflammatory cytokines released	[67]
		Male Swiss mice	CCI	10 mg/kg, i.p.	-	↓ Mechanical hyperalgesia	-	↓ Thermal hyperalgesia	-		-				-			-	↑ NO/cGMP/PKG/KATP channel signaling pathway, ↓ pro-inflammatory cytokines released	[68]
Verbascoside	Antinociceptive, Antioxidant,	Adult male Wistar rats	CCI	50, 100, and 200 mg/kg, i.p.		↓ mechanical allodynia	↓ heat hyperalgesia			↓ cold allodynia									↓ Oxidative stress	[69]
Hesperidin	Antinociceptive, antidiabetic	Male SD rats	CCI	50 mg/kg, i.p.	-	↓ Mechanical hyperalgesia	↓ Thermal hyperalgesia		-		-							-	↓ P2X3 receptor	[70]
		Male Wistar rats	CCI	100 mg/kg, i.p.	-	↓ Mechanical hyperalgesia	↓ Thermal hyperalgesia		-		-							-	↓ Pro-inflammatory cytokines released	[71]
		Adult SD rats	STZ-induced diabetic neuropathy	Hesperidin: 25, 50, 100 mg/kg, p.o. Insulin: 10 IU/kg, s.c.	↓ Mechanical hyperalgesia	↓ Mechano-tactile allodynia	-		↓ Thermal hyperalgesia		-							↑ MNCV, ↑ SNCV	↓ Glycated hemoglobin, AR activity, oxidonitrosative stress, neural calcium, and pro-inflammatory cytokines	[72]
Saikosaponin D	Antinociceptive	Male ICR mice	VCR-induced neuropathic pain	5,20 mg/kg, i.p.	-	↓ Mechanical hypersensitivity	-	-	-	-	-				-			-	↓ Activation of TRPA1 channel	[73]
Saikosaponin A	Antinociceptive, anti-inflammatory	Adult SD rats	CCI	6.25, 12.50 and 25.00 mg/kg, i.p.	-	↓ Mechanical allodynia	↓ Thermal hyperalgesia	-	-	-	-				-			-	↓ Activation of p38 MAPK and NF-κB signaling pathway	[55]
Paeoniflorin	Antinociceptive, anti-inflammatory	Male Wistar rats	CCI	50 mg/kg, i.p.	-	↓ Mechanical allodynia	↓ Thermal hyperalgesia	-	-	-	-				-			-	↓ Activation of p38 MAPK and NF-κB signaling pathway	[58]
		Male C57BL/6NCr mice	PTX-induced neuropathic pain	0.1 and 1%, applied topically (20 μL)	-	↓ Mechanical allodynia		-	-	-	-				-			-	↑ The activation of adenosine A1 receptor	[74]
		SD rats	CCI	Paeoniflorin: 25, 50, 100 mg/kg, i.p. MCC950:10 mg/kg, i.p. ML385:500 pmol/5 μL, i.t.	-	↓ Mechanical allodynia	↓ Thermal hyperalgesia	-	-	-	-				-			-	↓ Spinal NLRP3 infammasome activation	[75]
Morroniside	Antinociceptive	Adult male Wistar rats	SNL	30, 100, 300, 600 mg/kg, p.o. 3, 10, 30, 100, 300 μg, i.t.		↓ Mechanical allodynia	↓ Thermal hyperalgesia					No sedation or motor side effects							↑ Activation of GLP-1 receptor	[76]
		Wistar 1-day-old neonatal rat pups; male adult rat	SNL	300 μg, i.t.		↓ Mechanical allodynia													↑ GLP-1R/ (IL-10/β-Endorphin antinociceptive pathway.	[77]
liquiritin	Antinociceptive, anti-inflammatory, neuroprotective	ICR mice	CCI	30, 60, 120 mg/kg, i.p.		↓ Mechanical allodynia	↓ Thermal hyperalgesia				↓ Cold allodynia	→ Motor coordination	→ Exploratory behavior	↑ Sciatic function index				↑ MNCV, ↑ MNCP	↓ Pro-inflammatory cytokines, ↑ Anti- inflflammatory cytokine	[78]
Salidroside	Antinociceptive, anti-inflammation, stress reduce,	Lean rats	Zucker diabetic fatty (ZDF) rats	25, 50, 100 mg/kg, p.o.		↓ Mechanical hyperalgesia	↓ Thermal hyperalgesia											↑ SNCV	↓ Pro-inflammatory cytokines, ↓ P2X7 receptor	[61]
		Male SD rats	T2DM rat model	50, 100 mg/kg, p.o.	-	↓ Mechanical allodynia	↓ Thermal hyperalgesia												↑ AMPK activation, ↓ NLRP3 inflammasome activation	[79]
Isoorientin	Antinociceptive, Antioxidant,	Male ICR mice	CCI	7.5, 15, and 30 mg/kg/day, p.o.	-	↓ Mechanical allodynia	↓ thermal hyperalgesia		-		↓ cold allodynia							↑ SNCV, ↑ SNAP	↑ T-AOC, T-SOD, CAT, MDA ↓ Pro-inflammatory cytokines	[80]
Albiflorin	Antinociceptive, anti-anxiety and anti-depressant anti- inflammatory, neuroprotective,	Adult male SD rats	CCI	50 mg/kg, i.p.		↓ Mechanical allodynia									↓ Anxiety-like behavior	↓ Anxiety-like behavior	↓ Depression-like behavior		↓ NLRP3 inflammasome activity	[81]
Geniposide	Antinociceptive, anti-inflammatory,	Adult male SD rats	STZ-induced diabetic neuropathy	1, 10, 100 mg/kg, i.p.		↓ Mechanical allodynia													↓ Pro-inflammatory cytokines,	[82]

CCI: chronic constriction injury; SD: Sprague-Dawley; PTX: paclitaxel; STZ: streptozotocin; i.p.: intraperitoneal injection; i.t.: Intrathecal injection; p.o.: oral; →: Not affect/unchanged; ↑: Enhanced/Increased/Upregulate; ↓: Attenuate/Downregulate/Decrease/Suppress/Inhibit/Prevent.

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
