# Peer review of "Glycosides for Peripheral Neuropathic Pain: A Potential Medicinal Components"

_molecules, 2021, doi:10.3390/molecules27010255_

Round 1
Reviewer 1 Report
The authors prepared a well-written and well-organized manuscript aiming to summarize the epidemiology of neuropathic pain and the existing therapeutic drugs used for disease prevention and treatment. In the present manuscript, the authors suggested the pharmacological application of glycosides on neuropathic pain based on laboratory findings to improve patients’ life quality.
The manuscript is well-written and well-prepared. I found that the topic of current MS is adequate and acceptable for the journal’s scope. There is a novelty, and it is definitely worthy of publication. However, several concerns/questions, as mentioned below, need to be addressed by the authors before being considered for publication in this journal.
As the authors pointed out in Table 2 of their manuscript, glycosides isolated from medicinal plants have many pharmacological effects. For example, in addition to its anti-inflammatory effect, Diosmin also exerts antioxidant, antidiabetic, and anxiolytic effects. Another example, Paeoniflorin also has diverse pharmacological effects, not only acting as a neuroprotective agent. How glycosides can exert such diverse pharmacological effects and whether its analgesic effect can dominate other effects remain unclear in this manuscript. Please discuss this further.
The authors proposed that glycosides can be used for their analgesic effect. In the section on Conclusion and future perspectives, I recommend the authors include their perspectives on how glycosides can be used clinically in the future. For example, the authors can include steps that shall be passed in the drug discovery to have glycosides products to be available in the market.
Finally, I recommend that this study shall be considered for publication in this journal after addressing the comments/concerns above.
Reviewer 2 Report
The review must be improved, in particular some sections need to be revised.
Major points:
- In some sections, experiments are described in unnecessary detail (P4, Lines 137-144; P7, Lines 215-217), and in some briefly. In my opinion, there should not be excessively detailed information in the review (such as statistical data, isolation conditions etc.), as in experimental articles, but only a general description, interesting conclusions or what distinguishes this study from others.
- Section 4. “Synopsis of glycosides”: one of the glycoside classifications based on glycoside bonds is a common and well-known classification, so please rephrase this sentence (P. 4, Line 178).
Reference to Table 2 is out of the place.
It is not entirely clear why these particular groups of glycosides were chosen for the detailed description. Only these groups have been shown to be active in neuropathic pain models? Are all groups active ingredients of Chinese herbs? If yes, please, write it more specifically in this section.
- Section 5 it is better to rename ”The antinociceptive effects of glycosides in a rodent model of neuropathic pain”.
There are 11 groups of glycosides in the Table 2. However, group Geniposide is missed in the section 5.
- Section 6.3.2. Members of the transient receptor potential (TRP) channel family are known to be expressed in pain receptors and have attracted significant attention in the search for new therapies for neuropathic pain. Maybe you meant they are expressed in nociceptors?
Minor points:
P2. Lines 61, 63. The comma between two sentences should be replaced with a period.
treat neuropathic pain have emerged from epilepsy or neurology, Indeed, almost half of
alin and amitriptyline, Unfortunately, these drugs are often ineffective or induce intoler-
P2. Lines 71, 72. Glycosides
We also aimed to summarize the therapeutic effects of glycoside,
P5. Line 61, 63. The word «O-glycoside» is mentioned twice
glycoside, O-glycoside, S-glycoside, C-glycoside (Table 2), and O-glycoside (the most
P6. Line 206. Check the meaning. As usual, it may considered as? clinically significance as a 30% reduction in pain [95].
As usual, it may considered has clinically significance as a 30% reduction in pain [95].
P7. Line 222. The comma between sentence should be replaced with a period.
model of painful diabetic neuropathy for four weeks, Mice were then subjected to the
P7. Line 230. The sentence must start with a capital letter.
pendent manner [54]. saikosaponin D was also shown to significantly attenuate agonist-
P8. Lines 292-293. Italics.
Lippia citriodora H.B.K.
P9. Lines 252-256. Remove numbering, first, second, etc.
P10. Line 391. Add ion channels.
These neurons express different receptors and channels and respond to mediators
P11. Lines 408, 410. Add and.
Liquitin, saikosaponins, hesperidin
Paeoniflorin, albiflorin
P11. Lines 425, 429, 431. Too many word Researchers in this part.
P12. Lines 443, 450, 454. P2X7R, the clarification is already at the beginning of the section, you can remove.
P15. Line 592. TRPA1 is the one of the main…

Reviewer 3 Report
Peer-review of the article: Glycosides in the Treatment of Peripheral Neuropathic Pain, by Miao-Miao Tian et al.
Tian MM and collaborators prepared a review article on the effects of glycosides in animal models of peripheral neuropathic pain and suggested mechanisms of antinociceptive action as well. In the introduction section, they raised some concerns about the lack of diagnostic criteria and the quality of epidemiological studies undertaken to determine neuropathic pain prevalence and/or incidence. In general, the manuscript addresses an interesting topic, that falls within the scope of the journal, but I have very fundamental doubts related to the understanding of basic pharmacological and pain research concepts which are crucial for the critical literature reviewing process. Thus, my decision is detailed revision and re-submission.
I hope the following suggestions may aid in the improvement of the manuscript.
- Title: Avoid using the term „treatment“ in the title, because only preclinical studies are reviewed and no glycoside is used in the treatment of neuropathic pain in humans (at least not officially)
- Abstract:
- Statements about the efficacy of currently available medicines for neuropathic pain relief should be revised here and throughout the manuscript, because they are effective, maybe not in every single patient, so „limited efficacy and/or safety“ is more appropriate (Page 1, lines 13 and 14)
- Avoid using colloquialism „worse than death“ (Page 1, line 15)
- „by regulating oxidative stress, transcriptional regulation, ion channels, membrane receptors“…- this sentence is not complete (Page 1, lines 21 and 22)
- The term „antinociceptive“ is preferred as you are talking about preclinical studies (Page 1, line 24)
3. Introduction:
- „pain intensity, emotion…“ are incorrectly included as risk factors." (Page 1, lines 45 and 46)
- „from epilepsy or neurology“ – epilepsy is a neurological illness, so perhaps the authors meant "psychiatric" (Page 2, line 61)
- Authors point out that almost half of the currently available analgesics were developed for some other uses => antiepileptics are expected to be effective in the treatment of neuropathic pain since epilepsy and neuropathic pain share the same pathophysiology, namely neural hyperexcitability; in addition, drug repurposing is a growing practice in human medicine (Page 2, lines 60-63)
- Plant medicines, including glycosides, may have intolerable side effects similar to antiepileptic drugs/SNRIs/opioids (recall the cardiotonic glycosides safety profile) (Page 2, lines 63-65)
- The study's objectives are not well defined. The authors discuss therapeutic efficacy, which is not applicable in preclinical settings. The overall goal is not achievable because a clinical evaluation is required first. (Page 2, lines 70-75)
- Pharmacological treatment of neuropathic pain:
- Although analgesics are symptomatic rather than causal pain treatment options, this does not rule out the possibility that they have no impact on underlying pain mechanisms; in fact, all suggested mechanisms for glycosides-induced antinociceptive effects are commonly associated with pregabalin, gabapentin, duloxetine, and others. (Page 4, lines 145-151)
- Plant medicines’ safety profile warrants at least the same level of concerns as synthetic drugs (Page 4, lines 162-163)
- A very incorrect concluding sentence that „glycosides are widely used in the treatment of neuropathic pain (Page 4, lines 165-168)
- Synopsis of glycosides:
- One could assume from Table 2 that the mentioned glycosides have no antinociceptive properties. Why does the majority of glycosides lack this activity? (Pages 5 and 6)
- Antinociceptive effects of glycosides in mice models of neuropathic pain:
- This is the most critical part and is not well discussed!
- First, there are no inclusion and exclusion criteria for studies selection. It is interesting why studies performed on rats are not included, given how numerous they are.
- Second, the authors paraphrase the results of other authors not giving a critical analysis
- Finally, there is no conclusion at the end of the section about neuropathic pain states in which glycosides are especially effective as an implication for subsequent clinical evaluation.
- Immediate issues:
- The terms allodynia and hyperalgesia are not defined properly; furthermore, the distinction between mechanical and thermal hypersensitivity is not explained (Page 6, lines 196-201)
- Tabelar view (model, nociceptive test(s), route of administration, dose, primary and secondary outcomes, with comments if applicable) is more appropriate for the reader's eye
- The same approach in discussing others’ results should be given throughout the manuscript
- Re-phrase the conclusion (Page 8, lines 289-290), meaning that 7 days are needed for antinociceptive action development enough to be detectable, not that long-term use of glycoside is tolerable
- Many references are incorrectly cited, such as 88 (Page 9, line 317), 66 (Page 9, line 321), 117 (Page 10, line 367), 8 and 9 (Page 12, line 440), thus careful check of all data source is mandatory
- In the Conclusion paragraph (Page 9, lines 349 and 350) avoid brave and subjective observations
- Antinociceptive mechanisms underlying the action of glycosides on neuropathic pain:
- The first paragraph does not serve as an introduction to the following text (Page 9, lines 352-358)
- Mechanism-based treatment is closely related to disease/cause-based treatments. By knowing how certain disease (e.g., diabetes) or other cause (e.g., oxaliplatin) induces neuropathic pain development, we can use further mechanism-based approach to obtain optimal analgesic(s)
- Conclusions and future perspectives:
- Epidemiological studies on neuropathic pain are not in the scope of this review, and should not be mentioned in the Conclusion section
- Rephrase the sentence: „the concept of pain treatment based on neural mechanisms is emerging“
- Concerns about the limited penetration into the CNS by glycosides, poor oral bioavailability, and instability should be more precisely stated, as should their questionable safety
- The last sentence of this review is just not an implication of the manuscript and should be omitted.
Round 2
Reviewer 2 Report
The article has been revised according to the reviewer's recommendations. The added table makes the overview information easier to understand. I have no more comments.